# Quality Retention of Fresh Tuna Stored Using Supercooling Technology

**DOI:** 10.3390/foods9101356

**Published:** 2020-09-24

**Authors:** Taiyoung Kang, Timothy Shafel, Dongyoung Lee, Chang Joo Lee, Seung Hyun Lee, Soojin Jun

**Affiliations:** 1Department of Molecular Biosciences and Bioengineering, University of Hawaii at Manoa, Honolulu, HI 96822, USA; taiyoung@hawaii.edu (T.K.); lee272@hawaii.edu (D.L.); 2Department of Human Nutrition, Food and Animal Sciences, University of Hawaii at Manoa, Honolulu, HI 96822, USA; shafelt@hotmail.com; 3Department of Food Science and Biotechnology, Wonkwang University, Iksan 54538, Korea; cjlee@wku.ac.kr; 4Department of Biosystems Machinery Engineering, Chungnam National University, 99 Daehak-ro, Yuseong-gu, Daejeon 34134, Korea; 5Department of Smart Agriculture Systems, Chungnam National University, 99 Daehak-ro, Yuseong-gu, Daejeon 34134, Korea

**Keywords:** tuna, supercooling, shelf-life, food preservation, quality assessment

## Abstract

The present study was focused on the investigation of physiochemical changes in tuna subjected to a novel supercooling preservation, which was assisted using a combination of pulsed electric fields (PEF) and oscillating magnetic fields (OMF). Fresh tuna fillets were stored without freezing at −3.2 °C for 8 days. The electrochemical impedance spectroscopy (EIS) parameter $P_{y}$ indicated that there was a significant difference between the frozen-thawed samples (36.3%) and fresh (46.6%) and supercooled (45.9%) samples, indicating that cell damage from ice crystal growth did not occur in the supercooled tuna sample. The microstructure observation and drip loss measurement further confirmed that the ice crystal damage was present in frozen tuna, whereas no cellular damage was found in the supercooled samples. The EIS proved its ability to distinguish between tuna samples that were frozen or chilled (i.e., refrigerated and supercooled) during storage; however, it was less sensitive in detecting the extent of spoilage. Instead, the K-value was used to evaluate tuna freshness, and the measured K-values of the refrigerated, supercooled, and frozen tuna samples after 8 days of storage were 74.3%, 26.4%, and 19.9%, respectively, suggesting that the supercooling treatment significantly preserved the tuna fillets fresh with the improved shelf-life when compared to conventional refrigeration.

## 1. Introduction

Refrigeration and freezing, known as cold storage, are the most popular approaches to preserve highly perishable foods such as fish. Refrigeration slows down the chemical reactions and the growth of microorganisms. However, at refrigeration temperatures, the shelf-life of foods is often limited, and spoilage leads to substantial food waste [1]. Freezing has been recognized as the most common process for long-term preservation of food due to lowered storage temperatures and the immobilization of liquid water, which significantly reduces the rate of quality deterioration. Nevertheless, ice crystallization during freezing will cause irreversible microstructural damage to the food matrix through volumetric expansion [2], and this disruptive change decreases the consumer’s perception on the overall food quality.

Supercooling is a novel food processing technique that preserves food below its initial freezing point without the phase transition (solidification). Supercooling provides an extended shelf-life while maintaining food quality without ice crystal damage [3]. However, it is not only challenging to create, but also to maintain the supercooled state in food products for a sufficiently long period of time, because the supercooled state is thermodynamically unstable, and ice nucleation randomly occurs due to the inherent nature, leading to non-reproducible results [4,5].

In recent years, oscillating electric and magnetic fields were utilized to prevent ice nucleation during the freezing process. It was mainly postulated that the electric and magnetic fields potentially disturb and rotate water molecules present in food, and eventually prevent the aggregation of the molecules [6]. As a novel method, Mok et al. (2017) demonstrated that a combination of pulsed electric fields (PEF) and oscillating magnetic fields (OMF) could suppress ice formation within chicken breasts and maintain the supercooled state at −7 °C [7]. Although there have been prior studies that demonstrated the potential of the supercooling storage over the conventional chill methods, the supercooling procedure still needs to be validated in terms of both food safety and quality perspectives. Furthermore, it is crucial to ensure that a decent supercooling preservation method utilizing electric and magnetic field strengths would not lead to any cellular injuries in food matrix during the storage.

Electrochemical impedance spectroscopy (EIS) is a method of analyzing the electrical properties of a material by applying to it an alternating current over a range of frequencies, and measuring the corresponding electrical output signal [8]. EIS can be a direct approach to access minute changes in food quality because the electrical impedance is highly sensitive to the cell membrane properties such as permeability inside the food product [9]. In the literature, EIS has shown promise in its ability to distinguish between fresh and frozen-thawed food products [10,11]. Furthermore, many attempts have been made to use EIS in evaluating changes in the freshness of meats during storage [12,13]. Therefore, it was hypothesized that EIS could be utilized to identify freshness and microstructural integrity changes that might occur during a novel supercooling preservation treatment of fresh tuna.

In this study, the quality of fresh tuna (*Thunnus albacares*) after 8 days of the supercooled storage was investigated to address the extent of ice crystal damage and the freshness after treatments. The results were compared to tuna samples preserved under refrigeration and freezing environments for 8 days because (1) it is recommended that raw tuna is be consumed fresh in the refrigerator for up to 2 days, and (2) this present study was designed as a proof of concept verifying that supercooling could extend the shelf-life of highly perishable food materials, i.e., fish fillets with minimal quality deterioration.

## 2. Materials and Methods

### 2.1. Sample Preparation

Fresh yellowfin tuna fillets (*Thunnus albacares*) were purchased at a retail store (Honolulu, HI, USA). The tuna samples were cut to be within 80 g ± 3 g (5 × 5 × 3.5 cm^3^) and placed in a 3D-printed sample holder (Figure 1a). The cut samples were wrapped with polyethylene film to avoid dehydration during experimentation. The samples were stored at the ambient temperature of 4 °C (refrigeration), −3.2 °C (supercooling), and −18 °C (freezing) for 8 days, respectively, and the frozen samples were thawed in the refrigerator (4 °C) for 24 h. Quality factor assessments were conducted on day 0 and day 8 in triplicate for each treatment.

### 2.2. PEF and OMF Supercooling Treatment

A custom-built supercooling system was fabricated to generate PEF and OMF. To support maintaining the supercooled state of tuna for an extended period of time, (1) PEF (3.5 V_rms_ at 20 kHz) was delivered via a set of parallel titanium electrodes in direct contact with the sample, and (2) OMF (75 mT at 1 Hz) was generated by two electromagnets, which were aligned with the orthogonal directions of PEF (Figure 1a). An insulated-gate bipolar transistor (IGBT) (IRAMX20UP60A, International Rectifier, El Segundo, CA, USA) based power supply was used to produce a bipolar square pulse for PEF and OMF.

The PEF duty cycles and durations were slightly modified from a previous study for the controlled cooling rate to achieve a stable supercooled state within the samples [7]. Three different duty cycles (phase 1: 80%, 5 min; phase 2: 50%, 2 min; phase 3: 20%, 2 min) were sequentially repeated for the PEF treatment, and the OMF was applied at the PEF phase 3 [7]. The supercooling chamber was placed within a chest freezer (HF50CM23NW, Haier, Qingdao, China), and the freezer temperature was controlled by a proportional-integral-derivative (PID) controller (D1S -2R-220, SESTOS Electronics H.K., Hong Kong). A data acquisition system (Agilent 39704A, Agilent Technologies, Inc., Santa Clara, CA, USA) was used to monitor temperatures using K-type thermocouples and electrical signals, including voltage and current values using a current transducer (Model 411/150, Pearson Electronics, Palo Alto, CA, USA).

### 2.3. Electrochemical Impedance Spectroscopy (EIS)

The EIS was performed using a μ-Autolab type III potentiostatic frequency response analyzer (FRA) and NOVA software version 1.6 (Metrohm Autolab USA Inc., Riverview, FL, USA). Working and counter electrodes made of two stainless steel hypodermic needles (0.70 × 25 mm, Terumo Medical Corp., Tokyo, Japan) along with a Ag/AgCl reference electrode (Beckman Coulter, Inc., Brea, CA, USA) was used to measure the impedance of the tuna samples for 50 frequencies over the range 1000 kHz to 1.9 MHz at an applied amplitude of 0.2 V. The spectrum of each sample was taken four times at different locations with the insertion of the needle electrodes into the sample at a depth of 1 cm. A Nyquist plot was produced for each run, and the Fricke model equivalent electrical circuit was applied to each Nyquist plot in this study [12]; the analysis procedure and equivalent circuit are shown in Figure 1b. The parameter $P_{y}$ was used for quantitative analysis of the Nyquist plot and calculated by obtaining the value of ${Z^{\prime }}_{0}$ and ${Z^{\prime }}_{\infty }$ from the electrical equivalent circuit. The $P_{y}$ was given by the following equation [13]:(1)$$P_{y}=\frac{{Z^{\prime }}_{0}-{Z^{\prime }}_{\infty }}{{Z^{\prime }}_{0}}\times 100$$
where ${Z^{\prime }}_{0}$ is the resistance where the cell membrane begins to polarize, and ${Z^{\prime }}_{\infty }$ is the resistance where the frequency has become sufficiently small that all capacitor characteristics disappear and Warburg impedance dominates.

### 2.4. Drip Loss

After the initial sample preparation, each sample’s weight was measured and preserved at different storage conditions (refrigeration, supercooling, and freezing) for 8 days. The samples were weighed again after each treatment (final weight), and the drip loss was evaluated with the following equation:(2)$$\mathrm{Drip}\;\mathrm{loss}\;\left(\%\right)=\frac{\mathrm{initial}\;\mathrm{weight}\;-\;\mathrm{final}\;\mathrm{weight}}{\mathrm{initial}\;\mathrm{weight}}\times 100$$

### 2.5. Microstructure Analysis

Tuna samples after preservation were cut into sections 2.5 mm in thickness and fixed for 24 h with 4% paraformaldehyde in 0.1 M sodium cacodylate (pH 7.4). The fixed samples were dehydrated in a graded ethanol series (70, 80, 95, and 100%) and embedded in paraffin. Then, the samples were sectioned at 10 µm using a microtome (Leica, SM 2000R, Nussloch, Germany) and stained with hematoxylin and eosin. The tissue sections were histologically analyzed with a light microscope (Olympus BX 51, Tokyo, Japan) equipped with a digital image capture system.

### 2.6. ATP-Related Compounds and K-Value

The ATP-related compounds adenosine triphosphate (ATP), adenosine diphosphate (ADP), adenosine monophosphate (AMP), inosine-5′-monophosphate (IMP), inosine (Ino), and hypoxanthine (Hx) were assayed by HPLC (High Performance Liquid Chromatography) as described by Ryder (1985) with slight modifications [14]. The fish sample extract was prepared by homogenizing 5 g the sample with 25 mL of chilled 0.6 M perchloric acid for approximately 1 min. The homogenate was then centrifuged at 3000× *g* for 10 min at a temperature of 4 °C. Then, 10 mL of supernatant was immediately neutralized to pH 6.5–6.8 with 1 M potassium hydroxide. After 30 min at 0 °C, the precipitate, potassium perchlorate, was removed by passing through filter paper (Whatman #4, 125 mm Ø, General Electric Company, Pittsburgh, PA, USA). The extracts were stored in a −18 °C freezer until further analysis was performed.

An high-performance liquid chromatograph (HPLC) system (Model 1100, Agilent Technologies, Inc., Santa Clara, CA, USA) equipped with an isocratic pump (G130A), a variable wavelength detector (G1314A) with a standard flow cell (10 mm path length, 14 µL volume, and 40 bar max pressure), and an injector valve (G1328A) of 20 µL loop capacity was used to analyze the ATP-related compounds in the fish sample extracts. Separation of the compounds was achieved on a reverse-phase C18 stainless steel column (Grace Vydac, 218MS53, 3.2 mm ID × 250 mm, Hesperia, CA, USA). An injection of 10 µL of sample extract into an isocratic mobile phase of 100 mM phosphate buffer (pH 4.2) was used at a flow rate of 1 mL/min, a constant back pressure of 156 bar, and a column temperature of 20 °C. The eluent was monitored at 254 nm with full scale deflection set at 0.2 absorbance units. All solutions were filtered through a 0.45 µm nylon aqueous syringe filter (VWR, Radnor, PA, USA) prior to injecting onto the column. The elution of all compounds was complete within 8 min of running time. The areas below the chromatographic peaks obtained from the fish sample extracts were used to calculate the concentration of the ATP-related compounds as compared to an external standard curve. Chromatographic data were analyzed using ChemStation software (Agilent Technologies, Inc., Santa Rosa, CA, USA). All solutions in the HPLC procedure were made with Milli Q purified deionized water (Millipore, Inc., Bedford, MA, USA). The concentrations of the ATP-related compounds from the HPLC results were used to calculate the K-value for the tuna samples after each treatment with the following equation [15]:(3)$$K\left(\%\right)=\frac{\left(\mathrm{Ino}+\mathrm{Hx}\right)}{\left(\mathrm{ATP}+\mathrm{ADP}+\mathrm{AMP}+\mathrm{IMP}+\mathrm{Ino}+\mathrm{Hx}\right)}\times 100$$

### 2.7. Statistical Analysis

The differences between the factors measured for fresh, refrigerated, frozen, and supercooled samples were analyzed using the analysis of variance (ANOVA) along with Tukey’s multiple range tests at a significance level of 95% (SPSS version 20, SPSS Inc., Chicago, IL, USA).

## 3. Results and Discussion

### 3.1. PEF and OMF Treatment for the Extension of Supercooling within Tuna

The time-temperature profiles of tuna samples preserved with refrigeration, supercooling, and freezing are presented in Figure 2. Ice nucleation occurred during the freezing process and it could be determined by monitoring the sample temperature. The tuna samples tested had a freezing point at around −1.2 °C (averaged value), and a freezing plateau due to the released latent heat for the phase transition was clearly observed in the frozen samples. In contrast, no release of latent heat was detected in the supercooled samples during the experimentation, indicating that the applied PEF and OMF inhibited ice nucleation and maintained the supercooled state within tuna for an extended period. It was hypothesized that an alternating current electric field (named PEF in this study) could inhibit ice nucleation by vibrating and displacing water molecules [16]. Furthermore, OMF would enhance the transition dipole moments and vibrational states of the water molecules [17], and affect the diamagnetic properties of water [18], suggesting that the applied OMF may provide a stabilized supercooled state while combining the PEF. With the support of the combined PEF and OMF treatment, tuna samples were preserved at −3.2 °C for 8 days (Figure 2).

### 3.2. Electrochemical Impedance Spectroscopy (EIS) Analysis

Figure 3 shows the representative Nyquist plots and calculated $P_{y}$ values of tuna samples stored with refrigeration, supercooling, and freezing for 8 days. There was a small but significant decrease in $P_{y}$ in the refrigerated samples (44.1%) from the fresh samples (46.6%). Furthermore, significant differences were found between the frozen-thawed samples (36.3%), and fresh and supercooled samples (45.9%). The decrease in impedance, as indicated by a decline in $P_{y}$, in the refrigerated tuna samples might be directly associated with microbial growth and enzymatic reactions in tuna as spoilage proceeded. It was suggested that impedance could be used as an indicator of the freshness of fish because with increasing spoilage caused by microorganisms or enzymes, the concentration of dissolved ionic metabolites would increase, which resulted in an increase in conductivity or a decrease in impedance [19]. Although there was a significant decrease in $P_{y}$ of the refrigerated tuna samples after the 8-day storage period, the $P_{y}$ difference between fresh and refrigerated samples was not large enough (5.3% decrease) to distinguish the extent of spoilage. This might be due to the simple approach to electrode design (i.e., two stainless steel needle-type electrodes in Figure 1c). This basic EIS setup to measuring electrochemical impedance will not account for measurement artifacts and electrode polarization, which leads to a decrease in sensitivity of the measurement [9]. Therefore, the development of a more sophisticated electrode setup would be needed for improving the sensitivity of the method to better distinguish between spoiled and unspoiled tuna. Meanwhile, the strong decrease in $P_{y}$ in the frozen-thawed samples may be attributed to the destruction of an important number of cellular membranes by ice crystal growth in the extracellular spaces [10,20]. On the other hand, the results from the comparison of $P_{y}$ between treatments clearly indicated that no ice had formed within tuna during the supercooling preservation, and the supercooled samples could be considered less spoiled than the refrigerated samples. Based on the observations of the electrochemical parameter $P_{y}$, it would be more useful in differentiating between unfrozen and frozen-thawed tuna samples rather than detecting the onset of spoilage in tuna during storage [21].

### 3.3. Microstructure Analysis

The cross-sectional microscopic images of tuna stored at the refrigeration, supercooling, and freezing are presented in Figure 4. The tissue damages could be seen in frozen-thawed samples, whereas the muscle fibers of the supercooled samples showed no visible differences in the degree of muscle damage compared to the refrigerated samples. Freezing could result in irreversible mechanical damage to fish tissue due to the ice crystal formation. In general, rapid freezing leads to smaller and uniform ice crystals, which cause less damage to the tissue structure, whereas slow freezing produces relatively larger and irregular ice crystals that cause greater tissue damage [22]. Thus, it was likely that the formation of ice crystals ruptured the cell membranes and changed the structure of muscle fibers during the freezing procedure due to the slow rate of freezing employed in this study. Similar to the previous findings [4], the structural damage caused by ice crystals could be completely avoided in the supercooled tuna samples, and this result would provide evidence that no ice had formed within the tuna, which could be supported by the EIS data. Furthermore, it was evident that the applied PEF and OMF did not cause any undesirable microstructural changes.

### 3.4. Drip Loss

The drip loss of tuna fillets stored with the refrigeration, supercooling, and freezing for 8 days is shown in Figure 5. The drip loss of frozen-thawed tuna samples was significantly greater than those of the refrigerated samples, and the excessive drip loss could be visually perceived, as shown in the inserted image in Figure 5. The greater amount of fluid loss from the frozen-thawed samples would result from the cell wall rupture and denaturation of the protein. Garthwaite (1997) suggested that the growth of ice crystals may not account for the cell damages in fish since the cell walls are supposed to be considered elastic to withstand the damage caused by ice crystals [23]. However, in the present study, the samples were slowly frozen (−18 °C), indicating that comparatively large ice crystals may be formed within the extracellular matrix, which might lead to cell wall rupture, as shown in the morphology analysis (Figure 4) [24]. Furthermore, the EIS parameter $P_{y}$ clearly indicated that the ice crystal formation caused damage to the cell membranes of tuna meat, resulting in the higher drip loss because the $P_{y}$ can assess the integrity of the cell membranes of meat samples [13]. The drip loss from the frozen-thawed tuna samples could also be explained through the protein denaturation and aggregation, which decreased the water-holding capacity of the myofibrillar proteins of tuna meat [25]. On the other hand, there was no significant difference between the refrigerated and supercooled samples, indicating that the OMF and PEF treatment suppressed ice nucleation throughout the storage period and would not adversely affect the structural integrity within the muscle fiber.

### 3.5. K-Value

The K-value has been useful in determining quality as it negatively correlates with the freshness of fish, and the increase in K-value is related to the degradation of ATP. Based on the K-value, tuna fillets could be categorized as food spoilage (greater than 50%) and sashimi grade (less than 20%) [26]. The measured K-value results and the HPLC chromatographs of a supercooled and a refrigerated sample are presented in Figure 6. The fresh tuna samples had a K-value of 6.75 ± 0.73%, and all samples had significantly different K-values after 8 days of their respective treatments compared to the fresh samples (*p* < 0.05). The refrigerated samples showed the greatest increase in K-value (74.29 ± 1.49%) and thus were considered as spoiled, while the frozen tuna samples would be regarded as suitable for sashimi grade (19.89 ± 0.75%). The supercooled samples had only a slightly higher K-value (26.4 ± 0.88%) than the cutoff for sashimi grade. It was reported that the K-value-storage temperature relationship of tuna followed the first-order reaction, indicating that the rate of the enzymatic reaction would be highly temperature dependent [27]. Thus, the relatively small increase in K-value from the frozen tuna samples could be explained by the effectiveness of low temperatures. The HPLC chromatographs of tuna samples stored at the refrigeration and supercooling (Figure 6b) clearly demonstrated their sizable differences in Hx concentration. The hypoxanthine concentrations in the refrigerated and supercooled samples were 114.54 ± 14.82 and 25.25 ± 0.61 μg/mL, respectively, suggesting that the increase in K-value in the supercooled tuna samples was attributed to the transformation of IMP to inosine, whereas Hx was dominant in the refrigerated samples. It is known that the ATP breakdown process will proceed naturally over storage; however, the formation of Hx from Ino can be favored by bacterial activity [28]. Furthermore, other researchers were able to directly correlate an increase in Hx concentration with loss of sensory quality in seafood [29]. The supercooling preservation based on the combination treatment of PEF and OMF was capable of extending the shelf-life of tuna for 8 days with no ice crystal formation and was suitable for the improved storage method over refrigeration.

## 4. Conclusions

In the present study, the effects of supercooled storage on the quality attributes of tuna were investigated. The combination treatment of PEF and OMF was applied to suppress ice nucleation, and the tuna fillets could be preserved in the supercooled state at −3.2 °C for 8 days. The application of EIS was rapidly able to distinguish qualitatively between tuna samples that had been frozen and those that had not been frozen. The changes in the parameter $P_{y}$ of the supercooled tuna samples indicated that no ice was formed during the storage, and this fact was further verified by the drip loss measurement and muscle fiber microstructure observation. However, the EIS setup used in this study came to be inappropriate to differentiate freshness levels between the spoiled and unspoiled tuna samples due to the simple electrode setup. The K-value showed large significant differences between samples, which clearly indicated spoilage of the refrigerated sample after 8 days of storage, whereas the supercooled sample was still categorized as fit for consumption by its K-value, demonstrating the outstanding supercooling preservation for highly perishable food materials. The overall results demonstrated that the supercooling preservation would be advantageous over conventional refrigeration without causing a significant decrease in textural integrity of the product, which was observed with freezing. For future studies, the microbiological aspects of safety in supercooled food products should be addressed as a means of extending the shelf life of perishable foods.

## Figures and Tables

**Figure 1 foods-09-01356-f001:**
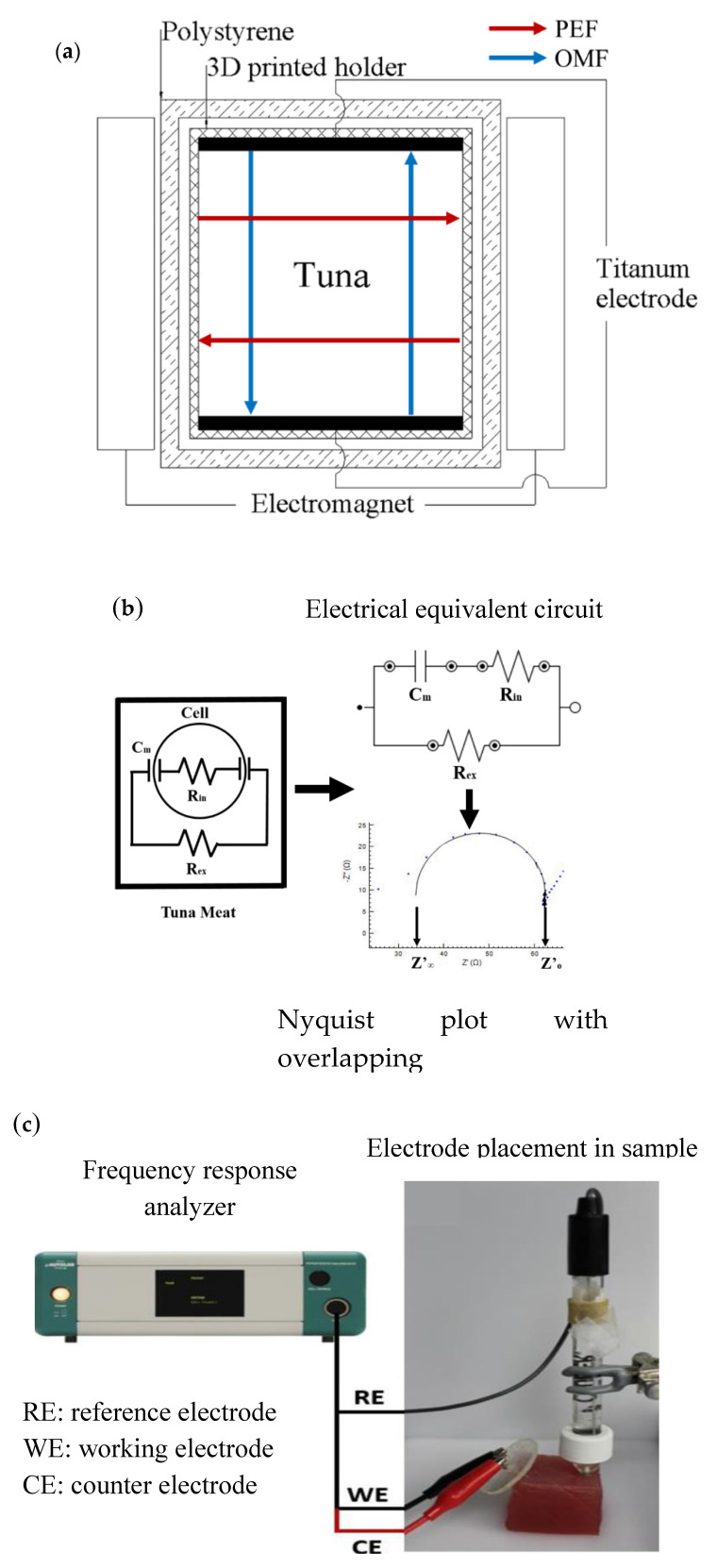
Schematic diagram of (**a**) the pulsed electric field (PEF) and oscillating magnetic field (OMF) supercooling system (arrows indicate the direction of the applied fields), (**b**) the electrical equivalence of tuna meat and its constituents (Fricke model), and the (**c**) electrochemical impedance spectroscopy (EIS) experiment setup.

**Figure 2 foods-09-01356-f002:**
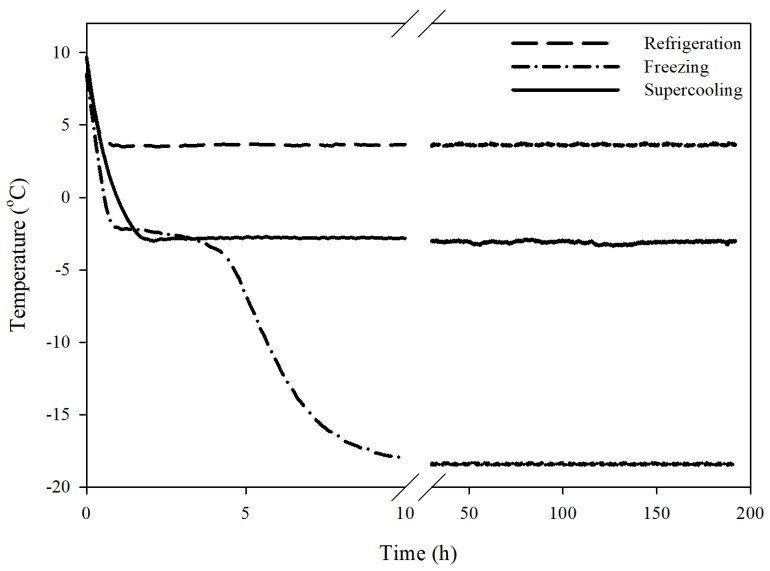
Representative temperature profile of the supercooled tuna (80 g) in comparison with a refrigerated (4 °C) and frozen (−8 °C) control sample. Note that freezing did not occur in the supercooled sample by the combined PEF and OMF treatment for 8 days at the temperature of −3.2 °C.

**Figure 3 foods-09-01356-f003:**
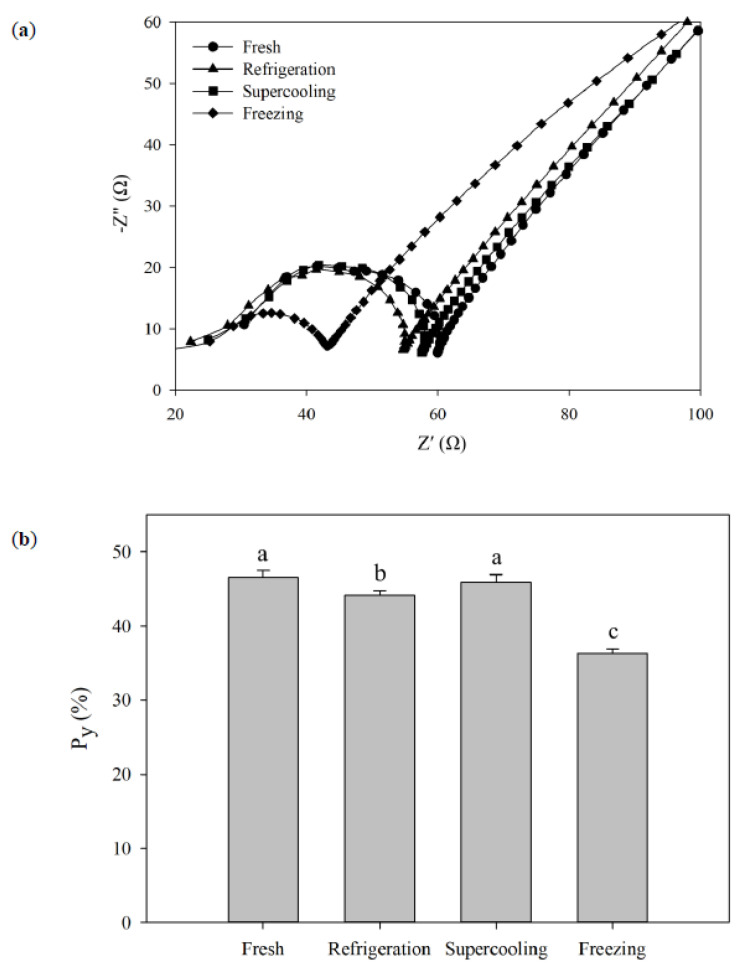
Nyquist plots (**a**) and the calculated $P_{y}$ values (**b**) of tuna (80 g) preserved at different storage conditions after 8 days. Error bars represent standard deviations, and different letters indicate significant differences (*p* < 0.05) between treatments.

**Figure 4 foods-09-01356-f004:**
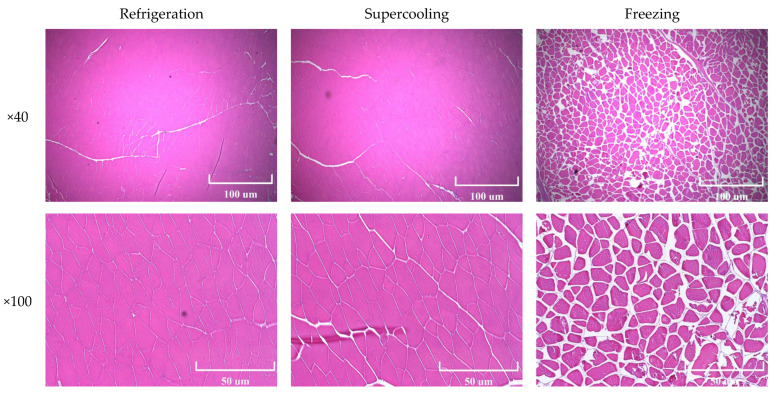
Optical microscopy images with different magnifications (×40 and ×100) of refrigerated, supercooled, and frozen-thawed samples.

**Figure 5 foods-09-01356-f005:**
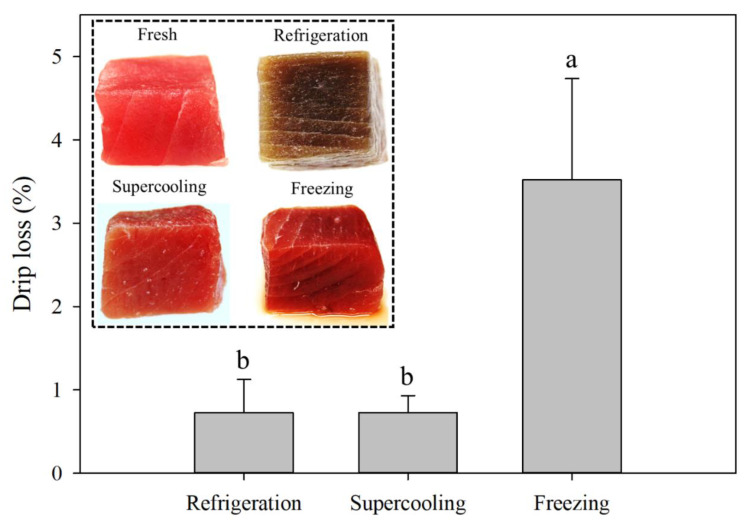
Mean vales of drip loss and the visual appearance (inserted figure) of tuna samples (80 g) stored for 8 days under the refrigeration, supercooling, and freezing. Error bars represent standard deviations, and different letters indicate significant differences (*p* < 0.05) between treatments.

**Figure 6 foods-09-01356-f006:**
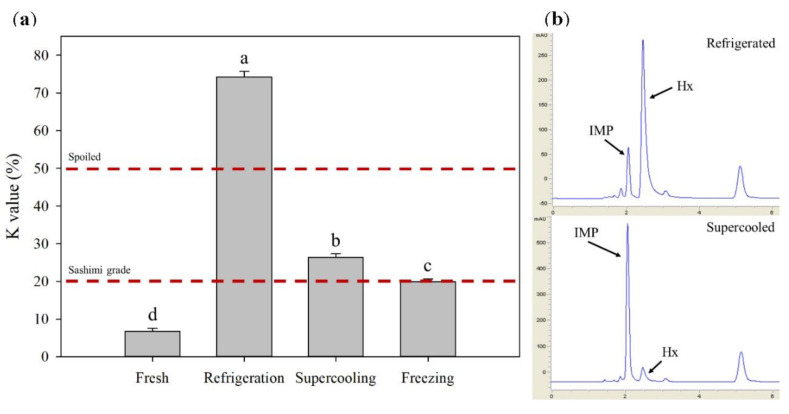
(**a**) K-values of sashimi-grade ahi tuna (80 g) after 8 days of respective treatments. between treatments. K-value cutoffs for spoiled (50%) and (20%) sashimi grade are given based on previous research [26]. Different letters indicate significant differences (*p* < 0.05) between treatments. (**b**) HPLC (High Performance Liquid Chromatography) chromatographs of tuna samples after refrigeration (8 days) (top) and PEF- and OMF-based supercooling (8 days) (bottom). Inosine5′-monophosphate (IMP) and hypoxanthine (Hx) peaks are indicated.

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
