# Peer review of "Quality Retention of Fresh Tuna Stored Using Supercooling Technology"

_foods, 2020, doi:10.3390/foods9101356_

Round 1
Reviewer 1 Report
Introduction
Please rewrite following sentences: 1. the first and second (lines 49-53); 2. The sentence starts with Nevertheless... (lines 55-59); 3. The sentence in lines 60-64. The sentence in lines 64-67.
They are used in already published papers, and may be considered as a plagiarism.
Materials and methods
2.1. Sample preparation
Please explain why did you choose this thawing method, and why did you do it till reaching core temperature of 20 oC?
Please describe the drip loss more in detail and cite the method if you used the previously described one.
Author Response
The authors appreciate the reviewer’s constructive comments and suggestions. The manuscript was revised to incorporate the review comments.
Introduction
Please rewrite following sentences: 1. the first and second (lines 49-53); 2. The sentence starts with Nevertheless... (lines 55-59); 3. The sentence in lines 60-64. The sentence in lines 64-67.
They are used in already published papers, and may be considered as a plagiarism.
Answer:
As suggested by the reviewer, the relevant sentences have been rewritten. Please see the revised manuscript (Line #49-53, #58-60, #61-64, and #70-77)
Materials and methods
2.1. Sample preparation
Please explain why did you choose this thawing method, and why did you do it till reaching core temperature of 20 oC?
Please describe the drip loss more in detail and cite the method if you used the previously described one.
Answer: In fact, the frozen tuna samples were thawed in a refrigerator (4°C ± 1°C) in this study. In Materials and methods, 2.1. Sample preparation and 2.4. Drip loss have been accordingly revised.
Reviewer 2 Report
The paper “Quality retention of fresh tuna stored using the supercooling technology” provides interesting results about the combined effect of pulsed electric fields (PEF) and oscillating magnetic fields (OMF) on a perishable food matrix. The experiments were well defined and the results are clear. Despite the number of analysed samples are limited, this could be considered as a proof of concept, as stressed by the authors. Other studies reported the effectiveness of these technologies on different meat matrices, however few data are reported on fish. Taking this in consideration, this article can provide a suitable contribution on this specific topic.
Broad comments
In general, I suggest to stress along the manuscript the innovation related to the seafood sector. I suggest to develop the discussion with a deepest comparison with available literature. Moreover, some details could be included in materials and methods.
Specific comments:
Abstract
Lines 33-34. there was a significant difference between the frozen-thawed samples (36.3%), and fresh 33 (46.6%) and supercooled (45.9%) samples.
These data are not well discussed/reported in the results section. Please, elucidate these data.
Introduction
Lines: 83-85, I suggest to rephrase these sentences.
Lines 97-98. I miss the sense of this sentence.
Materials and methods
Line 103: Sample preparation: retail store. Are there any specific information about the traceability of tuna fillets? Which assurance of the freshness status of fish was taken in consideration before the analyses?
Line 108: I suggest to rename sample class; I suggest to change along the manuscript Freezing with frozen/thawed. Why the authors did not test the behaviour of frozen/thawed samples during 8 days of refrigeration?
Line 109: samples were thawed at 20°C. did the authors analysed all the samples at room temperature?
Can the Samples Temperature affect the EIS analyses?
Line 110: triplicate. Did the author tested three different tuna fillets from different animals? Which factor (
Fish Size , season, fibre diameter .. ) can affect EIS measurements?
Results and discussion
Lines 244-247. In my opinion, the number of tested samples are limited to support these observations. Why the authors did not analysed a set of samples before the 8 days of conservation? The analyses of samples before the spoilage stage (e.g. at 4 days of conservation) may better elucidate the decay behaviour of each thesis (fresh vs supercooled vs frozen/thawed).
Lines 262. Samples could be classified. In general, classification performances are evaluated trough different statistical approaches. Moreover, the number of samples are limited to perform a strong cross validation. Authors can only observe a significant difference between samples.
Figure 3A. The Nyquist plots are poorly described and discussed.
Author Response
The authors appreciate the reviewer’s constructive comments and suggestions. The manuscript was revised to incorporate the review comments.
The paper “Quality retention of fresh tuna stored using the supercooling technology” provides interesting results about the combined effect of pulsed electric fields (PEF) and oscillating magnetic fields (OMF) on a perishable food matrix. The experiments were well defined, and the results are clear. Despite the number of analyzed samples are limited, this could be considered as a proof of concept, as stressed by the authors. Other studies reported the effectiveness of these technologies on different meat matrices, however few data are reported on fish. Taking this in consideration, this article can provide a suitable contribution on this specific topic.
Broad comments
In general, I suggest to stress along the manuscript the innovation related to the seafood sector. I suggest to develop the discussion with a deepest comparison with available literature. Moreover, some details could be included in materials and methods.
Specific comments:
Abstract
Lines 33-34. there was a significant difference between the frozen-thawed samples (36.3%), and fresh 33 (46.6%) and supercooled (45.9%) samples.
These data are not well discussed/reported in the results section. Please, elucidate these data.
Answer: As suggested by the reviewer, the manuscript has been revised (Line #262-265)
Introduction
Lines: 83-85, I suggest to rephrase these sentences.
Answer: The manuscript has been revised as suggested by the reviewer (Line #100-101)
Lines 97-98. I miss the sense of this sentence.
Answer: The manuscript has been revised as suggested by the reviewer (Line #115-116)
Materials and methods
Line 103: Sample preparation: retail store. Are there any specific information about the traceability of tuna fillets? Which assurance of the freshness status of fish was taken in consideration before the analyses?
Answer: The authors agree with the reviewer’s comment that points out the freshness of test samples. The retail store where tuna samples were purchased provided the information about “Processed date and time”. Our experimenters bought and supercooled samples at the same day.
Line 108: I suggest to rename sample class; I suggest to change along the manuscript Freezing with frozen/thawed. Why the authors did not test the behaviour of frozen/thawed samples during 8 days of refrigeration?
Answer: The authors addressed in multiple locations of the manuscript that the frozen samples were eventually thawed for quality factor assessments. Also as a post-freezing step, thawing itself has nothing to do with the quality deterioration to occur during the course of freezing. Therefore, the authors believe that the sample class of “freezing” would be simpler and less confusing to the readers. The target storage days were set as 8 days in this project, based on the stakeholders’ inputs. In future, the authors plan to check the quality degradation on a daily basis.
Line 109: samples were thawed at 20°C. did the authors analysed all the samples at room temperature?
Can the Samples Temperature affect the EIS analyses?
Answer: In fact, the frozen tuna samples were thawed in a refrigerator (4°C ± 1°C) in this study. The manuscript has been revised (Line #129). During the EIS analyses, all test samples (regardless of how they were stored) were kept in the refrigerator to obtain the stabilized preset condition.
Line 110: triplicate. Did the author tested three different tuna fillets from different animals? Which factor (
Fish Size , season, fibre diameter .. ) can affect EIS measurements?
Answer: The authors entailed three replicates for each experiment and three individual runs were performed. Various factors such as (sample size, electrode set-up, temperature, etc.) could affect the EIS measurements. Thus, the authors tried to keep experimental conditions consistent and repeatable, and the measurements were done in a triplicate for validity statistical analysis.
Results and discussion
Lines 244-247. In my opinion, the number of tested samples are limited to support these observations. Why the authors did not analysed a set of samples before the 8 days of conservation? The analyses of samples before the spoilage stage (e.g. at 4 days of conservation) may better elucidate the decay behaviour of each thesis (fresh vs supercooled vs frozen/thawed).
Answer: Please refer to our previous answer as above.
Lines 262. Samples could be classified. In general, classification performances are evaluated trough different statistical approaches. Moreover, the number of samples are limited to perform a strong cross validation. Authors can only observe a significant difference between samples.
Answer: As suggested the reviewer, the sentence has been revised (Line #286).
Figure 3A. The Nyquist plots are poorly described and discussed.
Answer: The authors agree with the comment raised by the reviewer. The parameters and obtained from the Nyquist plots are sensitive to the integrity of the cellular membrane and therefore can be a good indicator of cell damage. However, the parameters are highly dependent on the insertion depth of the electrodes. Instead, in this study, the electrochemical parameter Py was chosen to analyze the impedance spectra of the tuna samples. Normalizing the difference between and allows the sensing data to be more comprehensive, independent of electrode insertion depth (Pliquett, Altmann, Pliquett, & Schöberlein, 2003).
Added reference in the revised manuscript
Pliquett, U., Altmann, M., Pliquett, F., & Schöberlein, L. (2003). Py—a parameter for meat quality. Meat Science, 65(4), 1429–1437. http://doi.org/10.1016/S0309-1740(03)00066-4
Reviewer 3 Report
The study is interesting and it is well structured. Minor changes are requested to final acceptance.
Detailed comments
Check abstract, fish preparation and conclusions regards the temperature adopted in the supercooling process because two different temperatures were indicated (3,2 °C and 3,5°C)
Line 74: In the text was reported the year, but not the number. Check
Lines 127-130: How did you decide the three different duty cycles? Explain
Line 296: In the text was reported the year, but not the number. Check
Figure 6: “HPLC chromatographs of combination PEF and OMF supercooled (top) and refrigerated (bottom) tuna after treatments”. In the caption the supercooled and refrigerated positions have been reversed. Check
Some references are dated, is it possible to replace them with more recent ones?
Check the manuscript for grammatical language errors
Author Response
The authors appreciate the reviewer’s constructive comments and suggestions. The manuscript was revised to incorporate the review comments.
The study is interesting and it is well structured. Minor changes are requested to final acceptance.
Detailed comments
Check abstract, fish preparation and conclusions regards the temperature adopted in the supercooling process because two different temperatures were indicated (3,2 °C and 3,5°C)
Answer: The correction has been made in the revised paper (Line #127).
Line 74: In the text was reported the year, but not the number. Check
Answer: The reference number has been added as the reviewer suggested (Line #93).
Lines 127-130: How did you decide the three different duty cycles? Explain
Answer: As suggested by the reviewer, the authors have addressed the explanation why the different duty cycles and durations were used in Line # 148-150.
Line 296: In the text was reported the year, but not the number. Check
Answer: The reference has been added in the revised paper (Line #323).
Figure 6: “HPLC chromatographs of combination PEF and OMF supercooled (top) and refrigerated (bottom) tuna after treatments”. In the caption the supercooled and refrigerated positions have been reversed. Check
Answer: The corrections have been made in Figure 6 caption.
Some references are dated, is it possible to replace them with more recent ones?
Answer: As suggested by the reviewer, some references have been replaced in the revised paper (Line #452, Line #454, Line #506, Line #510, Line #513).
Check the manuscript for grammatical language errors
Answer: The grammatical errors were revisited for clarification and the language has been improved in the whole document.
Reviewer 4 Report
The manuscript entitled ‘Quality retention of fresh tuna stored using the supercooling technology’ presents a simple but interesting work with a good impact for seafood industry. Also, this is a well written manuscript. All sections have been designed and executed appropriately. So, there are not any suggestions to improve the manuscript.
Author Response
The authors appreciate the reviewer’s positiveness to support our manuscript.